# EXPLICIT CONNECTION DISTILLATION

## ABSTRACT

One effective way to ease the deployment of deep neural networks on resource constrained devices is Knowledge Distillation (KD), which boosts the accuracy of a low-capacity student model by mimicking the learnt information of a high-capacity teacher (either a single model or a multi-model ensemble). Although great progress has been attained on KD research, existing efforts are primarily invested to design better distillation losses by using soft logits or intermediate feature representations of the teacher as the extra supervision. In this paper, we present Explicit Connection Distillation (ECD), a new KD framework, which addresses the knowledge distillation problem in a novel perspective of bridging dense intermediate feature connections between a student network and its corresponding teacher generated automatically in the training, achieving knowledge transfer goal via direct cross-network layer-to-layer gradients propagation. ECD has two interdependent modules. In the first module, given a student network, an auxiliary teacher architecture is temporarily generated conditioned on strengthening feature representations of basic convolutions of the student network via replacing them with dynamic additive convolutions and keeping the other layers unchanged in structure. The teacher generated in this way guarantees its superior capacity and makes a perfect feature alignment (both in input and output dimensions) to the student at every convolutional layer. In the second module, dense feature connections between the aligned convolutional layers from the student to its auxiliary teacher are introduced, which allows explicit layer-to-layer gradients propagation from the teacher to the student via the merged model training from scratch. Intriguingly, as feature connection direction is one-way, all feature connections together with the auxiliary teacher merely exist during training phase. Experiments on popular image classification tasks validate the effectiveness of our method. Code will be made publicly available.

## 1 INTRODUCTION

Deep Neural Networks (DNNs) have achieved great success in tackling a variety of visual recognition tasks (Krizhevsky et al., 2012; Girshick et al., 2014; Long et al., 2015). Despite the appealing performance, the prevailing DNN models usually have large numbers of parameters, leading to heavy costs of memory and computation. Conventional techniques such as pruning weights from networks (Han et al., 2015; Li et al., 2017) and quantizing networks to use low-bit parameters (Courbariaux et al., 2015; Rastegari et al., 2016; Zhou et al., 2016) have proven to be effective for mitigating this computational burden. More recently, Knowledge Distillation (KD), another promising solution family to get compact yet accurate models, has attracted increasing attention.

The goal of KD is to transfer the learnt information (knowledge) of a high-capacity DNN model or an ensemble of multiple DNN models (teacher) to a low-capacity target DNN model (student), striking better accuracy-efficiency tradeoffs at runtime. There exist numerous KD methods to address the knowledge transfer from the teacher to the student. Many of them use a two-stage training process which begins with training a teacher model and keeping it fixed, and then learns a target student model by forcing it to match the outputted knowledge from the pre-trained teacher model. Various types of knowledge have been explored, such as outputted logits (Ba & Caruana, 2014; Hinton et al., 2015), intermediate feature representations (Romero et al., 2015; Zagoruyko & Komodakis, 2017), and relational information of model outputs or representations (Park et al., 2019; Tian et al., 2020). Instead of defining new types of knowledge, the other methods adopt one-stage training process,

jointly training the student and teacher/peer models using bidirectional knowledge distillation (Zhang et al., 2018a; Yao & Sun, 2020) or on-the-fly ensemble distillation (Lan et al., 2018; Anil et al., 2018) or multi-exit distillation (Phuong & Lampert, 2019). Both two-stage and one-stage KD methods as above typically treat the knowledge transfer as an optimization problem of formulating robust distillation loss functions via introducing more informative knowledge supervisions and more effective knowledge matching strategies.

Instead of extracting informative knowledge and designing alternative distillation loss functions to fit a desired knowledge transfer optimization objective as existing KD methods, we investigate a new technical perspective in this paper: casting the knowledge distillation problem into designing auxiliary connection paths between the student and teacher networks to enable explicit layer-to-layer gradients distillation from the teacher to the student via training them from scratch simultaneously. We are partially inspired by recent advances on DNN architecture engineering, which show that designing sophisticated feature connection paths such as residual connections (He et al., 2016) and dense connections (Huang et al., 2017) across neighboring layers can allow better information and gradients flow throughout a single network, making the training easy to have significantly improved performance in model accuracy and convergence. We conjecture this simple principle would also be crucial to open the door to develop a totally new knowledge distillation framework if we can merge the student and teacher into a single network temporarily during training phase, and can also separate them easily after training. To explore this hypothesis, we present Explicit Connection Distillation (ECD), a very simple knowledge distillation framework. Specifically, we decompose the design of ECD into two interdependent modules, namely auxiliary teacher generation and dense feature connection distillation. Recent works (Liu et al., 2020; Yue et al., 2020) show that searching a good alignment of structural feature channels between the student and teacher networks can bring improved knowledge distillation performance under the premise that a pre-trained teacher model is available. Regarding the first module of ECD, we hope the generated teacher can make a perfect structure alignment (both in input and output feature dimensions of every convolutional layer) to the student network in an easier manner (no need of time-consuming searching procedure as used in (Liu et al., 2020; Yue et al., 2020)) while can attain superior model capacity. To this goal, we retain all structural units of the student network in constructing the teacher architecture, except for replacing original convolutions by dynamic additive convolutions (Yang et al., 2019) which have proven to be very effective for enhancing model capacity in network architecture engineering research. Regarding the second module of ECD, we hope knowledge distillation can be realized through the explicit layer-to-layer flows of gradients from the teacher to the student instead of the conventional mimicking procedure. To this goal, we add dense feature connections from the convolutional layers of the student to those aligned layers of the auxiliary teacher, and train the merged model from scratch. After training, all feature connections can be naturally removed as their connection direction is only from the student to its auxiliary teacher which also exists merely in training phase.

Beyond the common wisdom of knowledge distillation that requires the knowledge mimicking process between the student and teacher networks, our ECD sheds new insight: by considering knowledge distillation from a novel student-to-teacher merging, co-training and splitting viewpoint, direct feature connections from the student to its well-aligned auxiliary teacher (generated from the student in an automatic manner) can also achieve competitive performance to improve low-capacity student models, as validated by extensive experiments on CIFAR-100 and ImageNet datasets.

## 2 RELATED WORK

In this section, we make a brief summary of existing knowledge distillation works.

**Two-stage KD methods**. The idea of training a compact model to mimic the functions learnt by a larger ensemble of models is firstly proposed in (Bucilă et al., 2006). Ba & Caruana (2014) extends this idea, showing that shallower yet wider neural networks can also approximate the functions previously learnt by deep ones. Hinton et al. (2015) presents the famous Knowledge Distillation (KD) method, which adopts a teacher-student framework for transferring learnt soft knowledge from a high-capacity teacher to a low-capacity target student network. In this framework, the teacher is pre-trained and fixed, and then its soft logits on the training data are used as the extra supervision to guide the training of the student besides the ground truth labels. FitNets (Romero et al., 2015) shows the intermediate feature representations learnt by the teacher can be used as the complementary

knowledge to soft logits, enhancing knowledge distillation performance to some extent. Zagoruyko & Komodakis (2017) proposes to use spatial attention activations instead of intermediate feature maps. Recent works further show the relations (Park et al., 2019) and higher order dependencies (Tian et al., 2020) of learnt logits or intermediate feature representations by the teacher capture important structural information, which are more effective knowledge to augment the knowledge distillation process. Following the teacher-student framework as above, many other works such as (Yim et al., 2017; Lee et al., 2018; Kim et al., 2018) attempt to further improve knowledge representations and matching losses.

**One-stage KD methods**. Unlike two-stage KD methods that rely on a pre-trained teacher model and one-way knowledge transfer from the teacher to the student, one-stage KD methods simplifying the knowledge distillation process by training all models simultaneously. Deep mutual learning (DML) (Zhang et al., 2018a) considers the teacher as a peer of the target student, showing that the outputted logits of the student can be used to assist the training of the teacher by a peer-teaching strategy. In (Yao & Sun, 2020), the authors further show that deep mutual learning can also be boosted by intermediate soft outputs, just like that in two-stage KD methods. Guo et al. (2020) follows the basic framework of DML but replaces the mutual distillation (student-to-student) by the ensemble (over students)-to-students distillation. ONE (Lan et al., 2018) presents an on-the-fly ensemble distillation method in which a stronger ensemble teacher learnt over a multi-branch network is used to enhance the training of every branch. Anil et al. (2018) extends this idea to accelerate large-scale distributed neural network training applications. Wu & Gong (2020) follows the basic framework of ONE but jointly uses the mutual distillation (student-to-student) and the ensemble-to-students distillation. Other improved variants include but are not limited to (Phuong & Lampert, 2019; Malinin et al., 2020).

**Other KD variants**. Besides model compression, there also exist a lot of works using knowledge distillation methodology to handle other applications. Self-distillation methods (Furlanello et al., 2018; Bagherinezhad et al., 2015) explore the benefits of KD techniques to improve the model training, assuming only a single network is available. Lopes et al. (2015) and Chen et al. (2019) address data-free knowledge distillation where the original training data are no longer accessible due to safety or privacy concerns. Goldblum et al. (2020) and Chung et al. (2020) extend the idea of KD to study how adversarial robustness of the teacher model can be transferred to the student. Li & Hoiem (2016) and Hou et al. (2018) combine KD with fine-tuning and retrospection to handle life-long learning scenarios. Adapting KD to other tasks such as multi-modal visual recognition and natural language processing is explored in (Garcia et al., 2018; Kim & Rush, 2016).

## 3 EXPLICIT CONNECTION DISTILLATION

In this section, we describe the formulation of the proposed Explicit Connection Distillation (ECD) framework, and detail how to design and implement its two key components: auxiliary teacher generation and dense feature connection distillation.

### 3.1 CONVENTIONAL KD FORMULATION

For a better understanding of our method, we start with the formulation of conventional KD methods. Let $S$, $T$ and $\theta_S$, $\theta_T$ denote a target student model, its teacher (either a single model or an ensemble of multiple models) and their parameters correspondingly, and let $x$ denote training data, and let $\mathcal{Q}$ be a set of layer location pairs where knowledge mimicking process between two networks is introduced. Regarding one-stage KD methods, where the student and teacher models are trained jointly, the overall objective function to be optimized can be defined as

$$\mathcal{L}_{\text{KD}} = \mathcal{L}_{\text{CE}}(\theta_S, x) + \mathcal{L}_{\text{CE}}(\theta_T, x) + \lambda \sum_{\boldsymbol{q} \in \mathcal{Q}} d(f_S^q(\boldsymbol{x}), f_T^q(\boldsymbol{x})), \quad (1)$$

where $\mathcal{L}_{\text{CE}}(\theta_S, x)$ and $\mathcal{L}_{\text{CE}}(\theta_T, x)$ are the standard cross-entropy loss functions of the student model and the teacher model, respectively. $d$ is the loss for knowledge mimicking, which measures the distance of learnt knowledge $f_S^q(\boldsymbol{x})$, $f_T^q(\boldsymbol{x})$ at all specific layer location pairs between the student and teacher models. As we discussed in introduction section, intermediate feature representations,

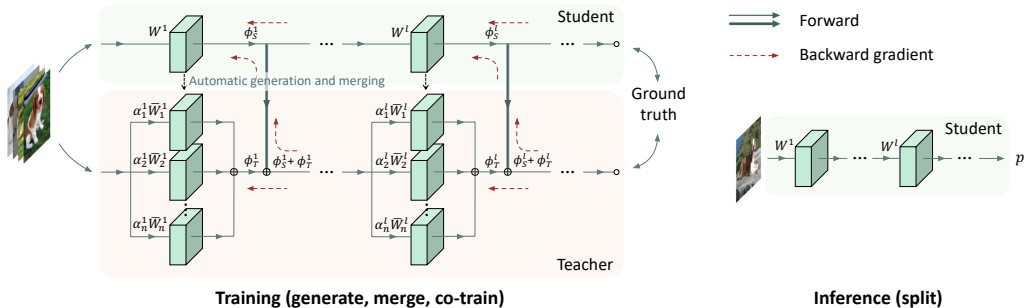

Figure 1: A schematic overview of the training stage and inference stage of our ECD. In the training phase, the teacher network is automatically generated by taking the student as the reference, and then both networks are merged by adding dense feature connections and co-trained from scratch. In the inference phase, in order to obtain the prediction ($p$), only the student network is needed which is simply split from the teacher.

soft logits, and their relations from learnt models are popularly used as the knowledge. $\lambda$ is a tunable weighting factor to balance loss terms, which is usually initialized to a relatively large value and decays during training. Simply removing the loss term $\mathcal{L}_{\mathrm{CE}}(\theta_T, x)$ from Equation (1) gets the objective function of two-stage KD methods, as in them the teacher model is pre-trained and fixed.

## 3.2 FORMULATION OF ECD

The goal of our ECD is to realize knowledge transfer from the teacher to the student without need of the common knowledge mimicking process described above, making KD design as simple as possible. That is, for ECD, the optimization objective is

$$\mathcal{L}_{\mathrm{ECD}} = \mathcal{L}_{\mathrm{CE}}(\theta_S, x) + \mathcal{L}_{\mathrm{CE}}(\theta_T, x). \tag{2}$$

As illustrated in Figure 1, merely given a student network and training data, our ECD achieves such a goal by a smart teacher-generating, student-to-teacher merging, co-training and splitting framework, consisting of two interdependent modules:

- **Auxiliary teacher generation module (in § 3.3)**: A teacher model, which structurally aligns with the given student at any network depth, is generated automatically from the student. Thus there is no longer any need to prepare large teacher architectures.

- **Dense feature connection distillation module (in § 3.4)**: The student model is temporarily merged with the teacher by adding explicit layer-to-layer connections to the teacher. The knowledge is transferred to the student during co-training of the merged model. Thus there is no longer any need to design distillation losses and tune $\lambda$ to balance different losses.

## 3.3 AUXILIARY TEACHER GENERATION

Unlike existing KD methods which usually need to prepare different teacher architectures for their corresponding student networks, in our method, a teacher model is generated automatically from the student by the auxiliary teacher generation module. To guarantee superior capacity of the generated teacher network and make a perfect structure alignment to the student network simultaneously, this module retains all structural units of any given student network in generating auxiliary teacher architecture, but replaces original convolutions by dynamic additive convolutions. This new type of convolution is recently proposed in (Yang et al., 2019), which has proven to be very effective for enhancing model capacity in network architecture engineering research. We extend it to construct the desired teacher which can naturally align with the student structure, and thus is compatible to our overall design goal. Denote convolutional kernels in the student network as $\mathcal{K}_S = \{\boldsymbol{W}^l\}, l \in 1, \cdots, L$, where $L$ is the number of convolutional layers. The generated kernels in the teacher network can be written as $\mathcal{K}_T = \{\sum_{i=1}^n \alpha_i^l \bar{\boldsymbol{W}}_i^l\}, l \in 1, \cdots, L$, where for the $l^{\mathrm{th}}$ layer and the $i^{\mathrm{th}}$ kernel $\bar{\boldsymbol{W}}_i^l$ it has the same shape with $\boldsymbol{W}^l$, $n$ is the kernel number, and we set it to 8/16 as suggested in (Yang et al., 2019), which will be also discussed in Table 13, and $\alpha_i^l$ is a learnable weight conditioned

on the input feature maps. Note that the set of $\alpha_i^l$ values are automatically learnt with a softmax function or a sigmoid function conditioned on the transformed features of the input channels. We experimentally used a softmax function, thus each $\alpha_i^l$ is within [0, 1] and their sum is equal to 1. Thanks to additive property, by replacing normal convolutions with dynamic additive convolutions, both input and output feature dimensions of every convolutional layer between the teacher and student networks are the same completely. You are referred to the block tagged as "teacher" in Figure 1 for a clear understanding of this generation process.

### 3.4 DENSE FEATURE CONNECTION DISTILLATION

Once the teacher and student networks are perfectly aligned at every convolutional layer, then effective knowledge distillation process of our method is realized by dense feature connections from the student to the teacher which merges two networks temporarily and enables co-training from scratch. As illustrated in Figure 1, we add skip connections to bridge both models, formulated as $\widehat{\phi}_T^l = \phi_T^l + \phi_S^l, l \in 1, \cdots, L$, where $\phi_S^l$ and $\phi_T^l$ are the feature maps after the $l^{th}$ convolutional layer of the student and teacher models, respectively, and $\widehat{\phi}_T^l$ is the new feature maps of the teacher after addition. We call this module dense feature connection distillation which is simple and neat, yet can well enhances the learning performance of the student model due to the backward gradient flows from the teacher. After training, all skip connections can be naturally removed as their connection direction is only from the student to its auxiliary teacher which also exists only in training phase.

### 3.5 UNDERSTANDING ECD

From the optimization objective of Equation 2, a common understanding of our method is the joint optimization will improve the training of the teacher as the student is merged into the teacher by the dense layer-to-layer feature aggregation progressively. It is true the teacher model is consistently improved as can be seen from the results in Table 16, but comparatively the improvement to the student model is usually larger. We can explain this in a reverse thinking: during the joint training, note that the student and the teacher are merged into one single network by dense layer-to-layer feature connections from the student to the teacher, that means during the inference the teacher will depend on the student but the student does not depend on the teacher (stripping away the teacher from the student). In such a perspective, by Equation 2, the teacher can be naturally treated as the auxiliary supervision to the student, which is well in line with the Deep Supervision (DS) methodology (Lee et al., 2015) in terms of both the mathematical formulation and the inference execution. In the DS methodology, it directly uses individual auxiliary supervisions added to several intermediate layers of a CNN model to ease gradients propagation, and they are discarded during inference. However, they usually bring marginal improvement on modern CNNs as reported (Huang et al., 2018; Zhang et al., 2018b; Sun & Yao, 2019). In a sharp contrast to existing DS methods, in our ECD, the teacher acting as the auxiliary supervision is not only based on its own structure generated with a perfect feature alignment (both in input and output dimensions) to the student at every convolutional layers, but also is based on the dense layer-to-layer feature aggregation from the student to the teacher progressively, enabling dense backward layer-to-layer gradients propagation from the teacher to the student and boosting the training of the student. Therefore, our method extends the deep supervision methodology in a new perspective on developing KD research.

## 4 EXPERIMENTS

In this section, we evaluate our method on CIFAR-100 and ImageNet classification datasets, and compare the performance against existing knowledge distillation methods. For fair comparisons, we use the public codes of different KD methods, and adopt the same training and data preprocessing settings throughout the experiments. All experiments are implemented with PyTorch. To our ECD, we add Conv1×1 on each connection path as we find it slightly improves the distillation performance (see ablative study part). Based on ECD, we also apply another ensemble distillation on the outputted logits from the two heads of the merged model to further boost the performance of each head, which is named as ECD*. Full implementation details are referred to Appendix.

Table 1: Main experimental results on the CIFAR-100 dataset. $^\dagger$ means using learning rate warm-up to smooth the training process. We report top-1 "mean (std)" accuracies (%) over 3 runs.

| Model | Student | Teacher | ECD | Gain | ECD* | Gain* |
|---|---|---|---|---|---|---|
| ResNet20 | 68.78 (0.22) | 71.05 (0.35) | 70.75 (0.29) | 1.97 | 71.07 (0.13) | 2.29 |
| ResNet32 | 70.80 (0.15) | 72.78 (0.44) | 72.40 (0.14) | 1.60 | 73.09 (0.14) | 2.29 |
| ResNet44 | 71.88 (0.13) | 73.48 (0.48) | 73.53 (0.16) | 1.65 | 74.08 (0.51) | 2.20 |
| ResNet56 | 72.29 (0.17) | 73.79 (0.51) | 73.94 (0.08) | 1.65 | 74.88 (0.05) | 2.59 |
| ResNet110 | 73.15 (0.45) | 75.32 (0.64) | 75.11 (0.14) | 1.96 | 75.47 (0.56) | 2.32 |
| ResNet164 | 76.64 (0.61) | 78.78 (0.73) | 77.94 (0.39) | 1.30 | 78.34 (0.79) | 1.70 |
| ResNet110$^\dagger$ | 74.41 (0.09) | 75.68 (0.21) | 75.59 (0.10) | 1.18 | 76.46 (0.20) | 2.05 |
| ResNet164$^\dagger$ | 77.15 (0.10) | 78.47 (0.11) | 78.33 (0.12) | 1.18 | 78.64 (0.19) | 1.49 |
| WRN-40-1 | 71.44 (0.14) | 72.48 (0.23) | 72.55 (0.17) | 1.11 | 72.83 (0.05) | 1.39 |
| WRN-40-2 | 75.96 (0.12) | 77.23 (0.19) | 76.55 (0.15) | 0.59 | 76.64 (0.07) | 0.68 |

## 4.1 EXPERIMENTS ON CIFAR-100

**Dataset**. CIFAR-100 (Krizhevsky & Hinton, 2009), containing 50,000 training images and 10,000 test images with 100 classes, is the most popular classification dataset for evaluating the performance of knowledge distillation methods.

**Implementation**. We performed experiments on prevalent ResNets (He et al., 2016) of different depths and WRNs (Zagoruyko & Komodakis, 2016) of different widths with the typical training settings (see Appendix for details). For auxiliary teacher models, we replace the standard convolutions in BasicBlock or Bottleneck of the student with dynamic additive convolutions (with $n = 16$ following (Yang et al., 2019)). For the experiments of each setting, we run each method 3 times and report top-1 "mean (std)" accuracies.

**Main results**. In Table 1, we provide average results for baselines, teachers and our method ECD. Note that results of baselines are slightly higher than those reported in the original papers. For our ECD experiments, we uniformly apply feature connections in the first two stages of ResNets or WRNs (we also provide ablative experiments to analyze where to add connection paths). Regarding to ResNet backbones, we observe $1.2\% \sim 2.0\%$ absolute accuracy gains for ECD, and $1.5\% \sim 2.6\%$ for ECD*. This shows that our design is effective for BasicBlock and Bottleneck structures with different depths. Besides, on WRN backbones, ECD and ECD* outperform baselines with $0.6\% \sim 1.1\%$ and $0.7\% \sim 1.4\%$ margins, respectively. In summary, the proposed ECD noticeably improves the performance of each student network, and ECD* further provides additional improvements. In most cases, the students trained by our method show even better performance than the corresponding teachers. These results verify the effectiveness of our proposed method. The performance is not that strong for WRN-40-2, as this architecture has already widened the network channels.

Table 2: Results comparison with state-of-the-art two-stage and one-stage KD methods. See Appendix for specific settings of these methods. We report top-1 "mean (std)" accuracies (%) over 3 runs.

| | Two-stage | | | | One-stage | | |
|---|---|---|---|---|---|---|---|
| Student | 68.78 (0.22) (ResNet20) | 70.80 (0.15) (ResNet32) | 71.44 (0.14) (WRN-40-1) | Student | 68.78 (0.22) (ResNet20) | 70.80 (0.15) (ResNet32) | 71.44 (0.14) (WRN-40-1) |
| Teacher | 73.15 (ResNet110) | 73.15 (ResNet110) | 73.23 (WRN-40-2) | Teacher | 74.43 (0.21) (ResNet110) | 75.27 (0.11) (ResNet110) | 73.99 (0.12) (WRN-40-2) |
| KD | 70.42 (0.16) | 72.08 (0.18) | 73.23 (0.23) | DML | 70.47 (0.25) | 72.57 (0.23) | 72.44 (0.21) |
| FitNet | 70.36 (0.16) | 72.09 (0.36) | 71.98 (0.19) | | | | |
| AT | 70.22 (0.03) | 71.64 (0.32) | 71.74 (0.10) | Teacher | 72.55 (0.18) (Ensemble) | 74.82 (0.08) (Ensemble) | N/A (Ensemble) |
| FSP | 69.95 (0.11) | 71.69 (0.06) | 71.61 (0.03) | | | | |
| SP | 70.35 (0.07) | 72.11 (0.32) | 72.47 (0.30) | | | | |
| VID | 70.14 (0.11) | 71.64 (0.30) | 71.68 (0.33) | ONE | 70.55 (0.07) | 72.29 (0.12) | N/A |
| PKT | 70.06 (0.08) | 71.66 (0.05) | 72.51 (0.18) | | | | |
| FT | 71.14 (0.37) | 72.82 (0.10) | 71.65 (0.09) | Teacher | 71.05 (0.35) (Generated) | 72.78 (0.44) (Generated) | 72.48 (0.23) (Generated) |
| NST | 70.12 (0.06) | 71.76 (0.36) | 71.31 (0.40) | | | | |
| RKD | 70.24 (0.09) | 71.93 (0.19) | 71.63 (0.21) | | | | |
| CC | 70.37 (0.19) | 71.67 (0.18) | 71.44 (0.09) | **ECD** | 70.75 (0.18) | 72.40 (0.14) | 72.55 (0.17) |
| CRD | 70.82 (0.12) | 72.84 (0.65) | 72.73 (0.18) | **ECD*** | 71.07 (0.12) | 73.09 (0.14) | 72.83 (0.05) |

Table 3: Results on the ImagNet dataset. We report top-1 accuracies (%).

| Model | Student | Teacher | ECD | Gain | ECD* | Gain* |
|-------|---------|---------|-----|------|------|-------|
| ResNet18 | 69.53 | 72.86 | 70.33 | 0.80 | 70.78 | 1.25 |
| ResNet50 (0.5×) | 71.41 | 75.27 | 72.19 | 0.78 | 72.48 | 1.07 |

**Comparison with SOTAs**. In Table 2, we compare our method with state-of-the-art knowledge distillation methods under the same training settings with ours. For ResNet20 and WRN-40-1, the automatically generated teachers of our method reach 71.05% and 72.48% accuracies (as previously shown in Table 1), which are much lower than the teachers used for other methods (73.15% and 76.23%), yet our method is able to achieve very competitive performance. For ResNet32, under the comparable capacity of the teach model (ours 72.78% vs. others 73.15%), our ECD* outperforms all the other methods with obvious margins.

## 4.2 EXPERIMENTS ON IMAGENET

**Dataset**. We additionally performed experiments on the ImageNet dataset (ILSVRC12) (Russakovsky et al., 2015), which is known as the most challenging image classification dataset containing about 1.2 million training images and 50 thousand validation images, and each image belongs to one of 1000 categories.

**Implementation**. Experiments are also conducted on prevalent ResNets. For auxiliary teacher models, we replace the standard convolutions in BasicBlock or Bottleneck of the student with dynamic additive convolutions (with $n = 8$ following (Yang et al., 2019)). We chose ResNet18 and Resnet50-half models, because they are often evaluated in knowledge distillation and the auxiliary teacher model and the student model have a suitable performance gap. Implementation details are available in Appendix A.1.

**Results**. Table 3 reports the performance of our method on ImageNet. ECD and ECD* improve baseline models by $0.8\%$ and over $1.0\%$ absolute gains in top-1 accuracy respectively, which support the benefit of our method on the large-scale dataset.

## 4.3 ABLATION STUDY

In this section, we isolate the influence of each element of our method, as well as compare with possible variants. All experiments are conducted on CIFAR-100 dataset. We use ECD but not ECD*, and do not use learning rate warm-up throughout all experiments for a better ablation study. For the experiments of each setting, we run our method 3 times and report top-1 "mean (std)" accuracies.

**Location of connections**. We explore the influence of the locations to add dense feature connections. This is very important because the semantics and robustness of different locations are different. We consider different settings by adding our dense feature connections to at most three blocks (including the block Conv2_$x$, Conv3_$x$, and Conv4_$x$, denoted as C2, C3 and C4 respectively) on CIFAR100. Detailed results are shown in Table 4. We observe that adding connections in both Conv2_$x$ and Conv3_$x$ brings the best performance improvement, followed by adding connections in Conv2_$x$ alone. These results indicate that adding connections in shallow layers such as Conv2_$x$ and Conv3_$x$ can migrate information well, while Conv4_$x$ extracts higher-level semantics and thus may be less

Table 4: Comparison of ECD with dense feature connections added to different blocks. In the table, C2, C3 and C4 refer to Conv2_$x$, Conv3_$x$ and Conv4_$x$, respectively. We report top-1 "mean (std)" accuracies (%) over 3 runs.

| Connection | ResNet20 | ResNet110 | WRN-40-1 |
|-----------|----------|-----------|----------|
| Baseline | 68.78 (0.22) | 73.15 (0.33) | 71.44 (0.14) |
| C2 | 70.36 (0.03) | 75.00 (0.35) | 72.00 (0.39) |
| C3 | 70.12 (0.12) | 74.41 (0.12) | 72.14 (0.38) |
| C4 | 69.75 (0.12) | 73.85 (0.60) | 71.32 (0.37) |
| C2+C3 | 70.75 (0.29) | 75.11 (0.14) | 72.55 (0.17) |
| C2+C3+C4 | 69.90 (0.10) | 73.75 (0.21) | 70.93 (0.18) |

Table 5: Results comparison of our ECD and using feature losses. "stage-wise" means feature supervision after each stage, which is consistent with many typical KD methods, and the distillation loss factor here is set to $10^{-2}$; "block-wise" refers to supervision on the same blocks where we add connections in our ECD, and the distillation loss factor here is set to $10^{-4}$. We report top-1 "mean (std)" accuracies (%) over 3 runs.

| Model | ResNet20 | ResNet110 | WRN-40-1 |
|---|---|---|---|
| Baseline | 68.78 (0.22) | 73.15 (0.45) | 71.44 (0.14) |
| l2-loss (stage wise) | 70.42 (0.16) | 74.64 (0.34) | 72.10 (0.22) |
| l2-loss (block wise) | 70.12 (0.10) | 74.32 (0.26) | 71.64 (0.15) |
| ECD | 70.75 (0.18) | 75.11 (0.12) | 72.55 (0.17) |

Table 6: Comparison with adding Conv1×1 on each connection path. We report top-1 "mean (std)" accuracies (%) over 3 runs.

| Transformation | ResNet20 | ResNet110 | WRN-40-1 |
|---|---|---|---|
| Baseline | 68.78 (0.22) | 73.15 (0.45) | 71.44 (0.14) |
| None | 70.42 (0.17) | 75.07 (0.21) | 72.13 (0.06) |
| Conv1×1 | 70.75 (0.18) | 75.11 (0.12) | 72.55 (0.17) |

robust, which limits the effect of connection supervision. To some extent, this is consistent with multi-task learning and multi-branch network designs (Lan et al., 2018; Anil et al., 2018) where the shallow network is shared and the higher level network is divided into separate branches.

**Comparison with using feature loss**. Recall that the distance of intermediate features between the student and teacher models is popularly used in existing KD methods to formulate feature loss (Romero et al., 2015; Zagoruyko & Komodakis, 2017). Instead of using feature loss, we simply use dense feature connections. In Table 5, we compare the performance of feature loss and dense feature connections under the same generated teacher. The results show that in our proposed framework, using dense connections is superior to using feature loss.

**Impact of the connection transformation**. In Table 6, we compare the performance when using direct connection and adding a Conv1×1 layer to each connection for improved alignment in feature semantics. We observe that adding Conv1×1 achieves slightly higher results than direct connection. That means direct connections without any transformation like Conv1×1 still get promising knowledge distillation performance. This is mainly due to our first module for auxiliary teacher generation. By replacing normal convolutions with dynamic additive convolutions, the teacher is well aligned to the student in network depth, and the input and output feature dimensions of every convolutional layer between the teacher and student networks are the same completely.

**Auxiliary teacher structure design**. As described in § 3.3, the teacher is generated automatically through replacing the convolutional kernels in the student by a linear combination of several kernels. As the knowledge is transferred from the generated teacher to the student, the capability of the teacher could affect the improvements. In Table 7, we compare our default setting (dynamic additive convolution, abbreviated as DAConv) with two other attention based alternatives SE (Hu et al., 2018) and CBAM (Woo et al., 2018). We find that under our proposed framework, generating teacher using attention modules such as SE and CBAM can also bring noticeable improvements over the student networks. Comparatively, our default setting achieves higher performance than these two alternatives.

**Embedding visualization**. To verify whether the student learns useful features during the ECD training, we provide visualization of T-SNE object embeddings in Figure 2. The figure illustrates that comparing with the baseline student model, applying ECD training helps learn more scattered embeddings, which provably affirms the advantage of our ECD.

**Combination experiment with DML and ED**. In this part, we compare our ECD with two typical response-based distillation methods including mutual distillation (DML) (Zhang et al., 2018a) and Ensemble Distillation (ED) (Lan et al., 2018). As shown in Table 8, our ECD with response-based distillation methods like DML and ED can achieve additional performance improvements. At the same time, we observe that the performance of the teacher model during ECD training is greatly improved as it is merged with student features. Under such stronger supervision of the teacher model,

Table 7: Comparison of auxiliary teacher generation with different methods. DAConv refers to our default setting. We report top-1 "mean (std)" accuracies (%) over 3 runs.

| Hyper-Conv | Method | ResNet20 | ResNet110 | WRN-40-1 |
|---|---|---|---|---|
| | Baseline | 68.78 (0.22) | 73.15 (0.45) | 71.44 (0.14) |
| SE | Teacher | 70.23 (0.33) | 74.98 (0.31) | 72.06 (0.23) |
| | ECD | 70.05 (0.17) | 74.96 (0.79) | 72.05 (0.17) |
| CBAM | Teacher | 70.45 (0.13) | 75.50 (0.11) | 72.26 (0.16) |
| | ECD | 70.08 (0.13) | 74.94 (0.59) | 72.26 (0.07) |
| DAConv | Teacher | 71.05 (0.35) | 75.32 (0.64) | 72.48 (0.23) |
| | ECD | 70.75 (0.29) | 75.11 (0.14) | 72.55 (0.17) |

Table 8: Combination experiment with DML and ED, where S represents the student model, T represents the teacher model and T+ represents the teacher model connected with the features of the student model under ECD training. For the training time of ResNet20 in the experiment, DML cost 5.40 gpu-hours and ECD cost 5.27 gpu-hours on a single NVIDIA TITAN RTX. Under the same teacher-student model, our ECD will be slightly faster than DML because ECD does not need to calculate additional loss.

| Model | Vanilla | | DML | | ED | | |
|---|---|---|---|---|---|---|---|
| | S | T | S | T | S | T | Ensemble |
| ResNet20 | 68.78 (0.22) | 71.05 (0.35) | 70.21 (0.08) | 72.96 (0.13) | 70.82 (0.19) | 73.14 (0.14) | 73.44 (0.18) |
| ResNet32 | 70.80 (0.15) | 72.78 (0.44) | 72.11 (0.07) | 74.91 (0.11) | 72.31 (0.13) | 75.16 (0.12) | 75.53 (0.12) |
| Model | ECD | | ECD+DML | | ECD+ED | | |
| | S | T+ | S | T+ | S | T+ | Ensemble |
| ResNet20 | 70.75 (0.29) | 72.40 (0.24) | 70.87 (0.14) | 73.01 (0.15) | 71.07 (0.13) | 73.41 (0.12) | 73.50 (0.16) |
| ResNet32 | 72.40 (0.14) | 74.83 (0.13) | 72.78 (0.16) | 75.08 (0.11) | 73.09 (0.14) | 75.28 (0.16) | 75.71 (0.07) |

the performance of student model is improved greatly, which is one of the reasons why our ECD can achieve better performance. In previous experiments, we mainly report the performance of the combination of ECD and ED, namely ECD*.

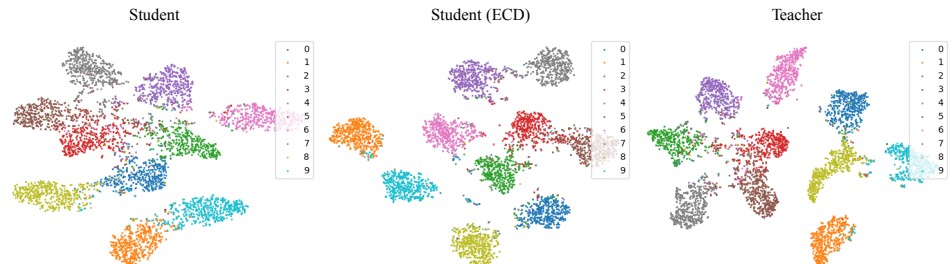

Figure 2: Visualization of object embeddings for the student, the student with ECD connections, and the teacher, by using T-SNE. Results are obtained based on CIFAR-10 with ResNet44.

In Appendix, we provide additional experiments, *e.g.* combining our method with labeling smoothing. More comparisons and discussions can also be found in Appendix.

## 5 CONCLUSION

In this paper, we present Explicit Connection Distillation (ECD), a new knowledge distillation framework, which addresses the knowledge distillation problem in a novel perspective of bridging dense intermediate feature connections between a student network and its corresponding teacher which is generated automatically in the training. ECD achieves knowledge transfer goal via direct cross-network layer-to-layer gradients propagation, without need to define distillation losses and assume a pre-trained teacher model to be available. We hope our work can inspire the future research on knowledge distillation designs.

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

# A APPENDIX

## A.1 MAIN EXPERIMENTAL SETTINGS

In this section, we provide detailed settings of the experiments conducted on the CIFAR-100 and ImageNet datasets.

### A.1.1 DETAILED EXPERIMENTAL SETTINGS ON CIFAR-100

The experiment was carried out on the CIFAR-100 dataset without additional data augmentation. The network is trained for 200 epochs, the batch size is 128, the weight decay is $5 \times 10^{-4}$, and the optimizer is SGD. For ResNets, we set the initial learning rate to 0.1, which decays by 0.1 at epochs 100 and 150. For WRNs, we do not adopt dropout, and the initial learning rate is set to 0.1, which decays by 0.1 at epochs 60, 120, and 160. The auxiliary teacher is expanded from $3 \times 3$ convolutional kernels (in the block of the original network) to dynamic additive convolutional kernels, and the number of multiplexing weight is 16. The Conv2_$x$ and Conv3_$x$ blocks of models have dense feature connections as shown in Table 9. Features after $3 \times 3$ convolution and after the block will be connected to the same position on the teacher network as shown in Figure 3. The features of the connected parts can be aligned better with the channel semantically with a little gain brought by $1 \times 1$ convolutions. For ECD*, a learnable ensemble is applied on both outputted logits of the student and the teacher.

### A.1.2 DETAILED EXPERIMENTAL SETTINGS ON IMAGENET

In the ImageNet experiments, the student model is trained with 100 training epochs, which is common setting[1]. The batch size is set to 256 and the multi-step learning rate is initialized to 0.1, which decays by 0.1 at 30, 60, and 90 epochs. The auxiliary teacher has the same ResNet structure except that

---

[1] https://github.com/pytorch/examples/tree/master/imagenet

the ordinary convolution is replaced with the dynamic additive convolution with the multiplexing weight 8. In the training process, the features of Conv2_x, Conv3_x and Conv4_x blocks of the student network are connected to the alignment positions of the auxiliary teacher enhanced by $1\times1$ convolution layers and BN layers. The output logits of the two output to supervise the training of the two branch heads. In the experiment, we chose ResNet18 and Resnet50-half as shown in Table 10 to verify the effectiveness of our method.

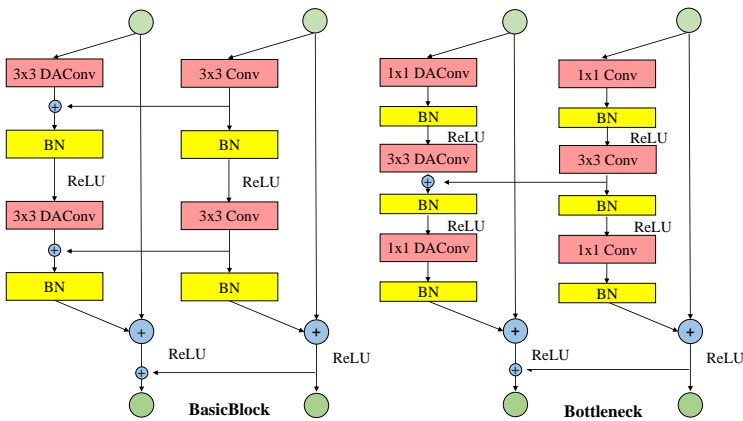

Figure 3: The detailed structure diagram of dense feature connections in BasicBlock and Bottleneck of ResNets, where DAConv refers to the dynamic additive convolution.

Table 9: Details of the convolutional blocks of the ResNet110, ResNet164 and WRN-40-2 backbones evaluated on the CIFAR-100 dataset.

| layer name | output size | ResNet110 | ResNet164 | WRN-40-2 |
|---|---|---|---|---|
| conv1 | 32×32 | 3×3, 16 | 3×3, 16 | 3×3, 16 |
| conv2_x | 32×32 | $\begin{bmatrix} 3 \times 3,\ 16 \\ 3 \times 3,\ 16 \end{bmatrix} \times 18$ | $\begin{bmatrix} 1 \times 1,\ 16 \\ 3 \times 3,\ 16 \\ 1 \times 1,\ 64 \end{bmatrix} \times 18$ | $\begin{bmatrix} 3 \times 3,\ 32 \\ 3 \times 3,\ 32 \end{bmatrix} \times 6$ |
| conv3_x | 16×16 | $\begin{bmatrix} 3 \times 3,\ 32 \\ 3 \times 3,\ 32 \end{bmatrix} \times 18$ | $\begin{bmatrix} 1 \times 1,\ 32 \\ 3 \times 3,\ 32 \\ 1 \times 1,\ 128 \end{bmatrix} \times 18$ | $\begin{bmatrix} 3 \times 3,\ 64 \\ 3 \times 3,\ 64 \end{bmatrix} \times 6$ |
| conv4_x | 8×8 | $\begin{bmatrix} 3 \times 3,\ 64 \\ 3 \times 3,\ 64 \end{bmatrix} \times 18$ | $\begin{bmatrix} 1 \times 1,\ 64 \\ 3 \times 3,\ 64 \\ 1 \times 1,\ 256 \end{bmatrix} \times 18$ | $\begin{bmatrix} 3 \times 3,\ 128 \\ 3 \times 3,\ 128 \end{bmatrix} \times 6$ |
| classifier | 1×1 | average pool, 100-d fc, softmax | | |

## A.2 COMPARISON EXPERIMENTS ON CIFAR-100

In order to make a fair comparison with existing knowledge distillation methods, we refer to the original settings and CRD code zoo[2] to implement various knowledge methods. The setting details, including settings of the factor ($\lambda$) in Equation (1), are shown in Table 11. We adopt the same standard training settings as our CIAFR-100 experiments in Table 1.

## A.3 MORE DISCUSSIONS ABOUT ECD

**ECD & label smoothing**. Label smoothing is an effective regularization method. Noise is added through soften one-hot labels, which reduces the weight of the real sample label category when calculating the loss function, and suppresses over-fitting. Table 12 indicates that ECD can be combined with label smoothing to provide additional improvements as well.

---

[2]https://github.com/HobbitLong/RepDistiller

Table 10: Structures of convolutional blocks of the ResNet backbones for the ImageNet dataset.

| layer name | output size | ResNet18 | ResNet50 (0.5x) |
|---|---|---|---|
| conv1 | 112×112 | 7×7, 64, stride | 7×7, 32, stride |
| | | 3×3 max pool, stride | |
| conv2_x | 56×56 | $\begin{bmatrix} 3 \times 3,\ 64 \\ 3 \times 3,\ 64 \end{bmatrix} \times 2$ | $\begin{bmatrix} 1 \times 1,\ 32 \\ 3 \times 3,\ 32 \\ 1 \times 1,\ 128 \end{bmatrix} \times 3$ |
| conv3_x | 28×28 | $\begin{bmatrix} 3 \times 3,\ 128 \\ 3 \times 3,\ 128 \end{bmatrix} \times 2$ | $\begin{bmatrix} 1 \times 1,\ 64 \\ 3 \times 3,\ 64 \\ 1 \times 1,\ 256 \end{bmatrix} \times 4$ |
| conv4_x | 16×16 | $\begin{bmatrix} 3 \times 3,\ 256 \\ 3 \times 3,\ 256 \end{bmatrix} \times 2$ | $\begin{bmatrix} 1 \times 1,\ 128 \\ 3 \times 3,\ 128 \\ 1 \times 1,\ 512 \end{bmatrix} \times 6$ |
| conv5_x | 8×8 | $\begin{bmatrix} 3 \times 3,\ 512 \\ 3 \times 3,\ 512 \end{bmatrix} \times 2$ | $\begin{bmatrix} 1 \times 1,\ 256 \\ 3 \times 3,\ 256 \\ 1 \times 1,\ 2048 \end{bmatrix} \times 3$ |
| classifier | 1×1 | average pool, 1000-d fc, softmax | |

Table 11: Brief implementation details of KD methods and settings of the distillation loss factor ($\lambda$).

| Method | factor ($\lambda$) | Brief notes of Implementation details. |
|---|---|---|
| KD (Hinton et al., 2015) | 0.1 | The temperature factor T is 4. |
| FitNet (Romero et al., 2015) | 0.1 | One stage without hint layer. |
| AT (Zagoruyko & Komodakis, 2017) | 1000 | The sum of absolute values with power p=2 is used as the attention. |
| FSP (Yim et al., 2017) | 1 | |
| SP (Tung & Mori, 2019) | 3000 | |
| VID (Ahn et al., 2019) | 1 | Hidden channel number is the same as output channel, and remove BN in $\mu$. |
| PKT (Passalis & Tefas, 2018) | 1000 | |
| FT (Kim et al., 2018) | 200 | |
| NST (Huang & Wang, 2017) | 10 | The polynomial kernel d is 2 and c is 0. |
| RKD (Park et al., 2019) | (25,50) | The distance is 50 and the angle is 25. |
| CC (Peng et al., 2019) | 100 | |
| CRD (Tian et al., 2020) | 0.8 | |
| DML (Zhang et al., 2018a) | 1 | |
| ONE (Lan et al., 2018) | 1 | Ensemble distillation of three branches. |

Table 12: Experiment results of applying Label Smoothing (LS) to ECD.

| Method | ResNet20 | ResNet110 | WRN-40-1 |
|---|---|---|---|
| Baseline | 68.78 (0.22) | 73.15 (0.45) | 71.44 (0.14) |
| Label smoothing | 69.89 (0.16) | 74.59 (0.31) | 71.56 (0.08) |
| ECD | 70.75 (0.29) | 75.11 | 72.55 (0.17) |
| ECD + LS | 71.03 (0.08) | 75.13 (0.43) | 72.66 (0.07) |

**Multiplexing weight settings**. In Table 13, we explore different settings of our ECD by changing the multiplexing weight for DAConv. We observe that the number 16 seems to be the best choice, but the distillation performance is not sensitive to the multiplexing weight (*e.g.* 72.12% v.s. 72.40% by doubling the number from 8 to 16). Besides, by simply setting the multiplexing weight to 2 or 4, we are still able to obtain obvious improvements.

Table 13: Experiment results of ResNet32 on CIFAR100 in different number of multiplexing weight for DAConv. We report top-1 "mean (std)" accuracies (%) over 3 runs

| | | Number of multiplexing weight | | | | |
|---|---|---|---|---|---|---|
| Model | Baseline | 2 | 4 | 8 | 16 | 32 |
| ResNet32 | 70.80 (0.15) | 71.47 (0.09) | 71.87 (0.16) | 72.12 (0.07) | 72.40 (0.14) | 71.86 (0.20) |

**More comparison experiments with other KD methods**. As shown in Table 14, the performance of our ECD and ECD* is quite good compared to other methods especially when combining with the original KD. Furthermore, we experiment with more dense cross-network "multi-layer-to-single-layer" feature connections in ECD, called ECD (dens), and the corresponding method combined with ED, called ECD (dens)*. Note that "multi-layer" indicates the student layer (at the same depth with the teacher layer) as well as its all previous layers. As shown in Table 14, our ECD (dens) and ECD (dens)* achieve the best performance among these KD methods, indicating that the feature connection in our ECD can effectively transmit feature distillation information. Also, we evaluate the performance of our ECD when combining with FitNet (Romero et al. (2015)) or CRD (Tian et al. (2020)), by adding FitNet loss or CRD loss during ECD training. We observe such combinations bring additional improvements, which means our method is orthogonal to other distillation methods such as FitNet and CRD. In Table 15, we refer to the results reported in CRD for two-stage KD methods, and we implement one-stage KD methods under the same training setting with CRD. As shown in Table 15, our ECD and ECD* achieve good performance, ECD (dens) and ECD (dens)* obtain the best performance for ResNet20 and ResNet32. Besides, we observe that the combination of ECD and other methods can achieve additional performance improvements.

Table 14: Results comparison with state-of-the-art two-stage KD methods and results of combining two-stage KD methods with the original KD. **Bold** and underline denote the best and the second best results, respectively. We report top-1 "mean (std)" accuracies (%) over 3 runs.

| Teacher | ResNet110 (73.15) | | ResNet110 (73.15) | | WRN-40-2 (76.23) | |
|---|---|---|---|---|---|---|
| Student | ResNet20 (68.78) | | ResNet32 (70.80) | | WRN-40-1 (71.53) | |
| Method | w/o KD | w/ KD | w/o KD | w/ KD | w/o KD | w/ KD |
| FitNet | 70.36 (0.16) | 70.46 (0.11) | 72.09 (0.36) | 72.70 (0.17) | 71.98 (0.19) | 73.14 (0.19) |
| AT | 70.22 (0.03) | 70.90 (0.21) | 71.64 (0.32) | 72.51 (0.19) | 71.74 (0.10) | 72.79 (0.38) |
| SP | 69.95 (0.11) | 70.87 (0.15) | 71.69 (0.06) | 72.92 (0.24) | 71.61 (0.03) | 72.77 (0.15) |
| CC | 70.35 (0.07) | 71.04 (0.31) | 72.11 (0.32) | 72.92 (0.19) | 72.47 (0.30) | 72.73 (0.37) |
| VID | 70.14 (0.11) | 70.71 (0.39) | 71.64 (0.30) | 72.81 (0.24) | 71.68 (0.33) | 72.68 (0.27) |
| RKD | 70.06 (0.08) | 70.71 (0.09) | 71.66 (0.05) | 73.51 (0.31) | 72.51 (0.18) | 72.51 (0.35) |
| PKT | 71.14 (0.37) | 71.01 (0.05) | 72.82 (0.10) | 72.88 (0.10) | 71.65 (0.09) | 72.93 (0.19) |
| FT | 70.12 (0.06) | 71.49 (0.05) | 71.76 (0.36) | 72.92 (0.24) | 71.31 (0.40) | 72.69 (0.03) |
| FSP | 70.24 (0.09) | 70.60 (0.07) | 71.93 (0.19) | 72.88 (0.15) | 71.63 (0.21) | 72.71 (0.13) |
| NST | 70.37 (0.19) | 70.94 (0.06) | 71.67 (0.18) | 72.96 (0.24) | 71.44 (0.09) | 72.79 (0.23) |
| CRD | 70.82 (0.12) | 71.26 (0.09) | 72.84 (0.65) | 73.18 (0.13) | 72.73 (0.18) | 73.54 (0.22) |
| ECD | 70.75 (0.18) | | 72.40 (0.14) | | 72.55 (0.17) | |
| ECD* | 71.07 (0.12) | | 73.09 (0.14) | | 72.83 (0.05) | |
| ECD+FitNet | 70.91 (0.16) | | 72.79 (0.21) | | 72.96 (0.12) | |
| ECD+CRD | 71.33 (0.09) | | 73.39 (0.11) | | 73.36 (0.16) | |
| ECD (dens) | 71.25 (0.13) | | 73.11 (0.17) | | 73.11 (0.21) | |
| ECD (dens)* | **72.44 (0.21)** | | **74.19 (0.12)** | | **73.76 (0.14)** | |

Table 15: Results comparison with state-of-the-art two-stage KD methods reported in CRD (Tian et al. (2020)) under the training setting of the 240 epochs. **Bold** and underline denote the best and the second best results, respectively.

| | Two-stage | | | | One-stage | | |
|---|---|---|---|---|---|---|---|
| Student | 69.06 (ResNet20) | 71.14 (ResNet32) | 71.98 (WRN-40-1) | Student | 69.06 (ResNet20) | 71.14 (ResNet32) | 71.98 (WRN-40-1) |
| Teacher | 74.31 (ResNet110) | 74.31 (ResNet110) | 75.61 (WRN-40-2) | Teacher | 74.68 (0.11) (ResNet110) | 75.53 (0.13) (ResNet110) | 75.74 (0.14) (WRN-40-2) |
| KD | 70.67 (0.27) | 73.08 (0.18) | 73.54 (0.20) | DML | 70.77 (0.20) | 72.87 (0.17) | 72.63 (0.15) |
| FitNet | 68.99 (0.27) | 71.06 (0.13) | 72.24 (0.24) | Teacher | 72.85 (0.16) (Ensemble) | 75.21 (0.17) (Ensemble) | N/A (Ensemble) |
| AT | 70.22 (0.16) | 72.31 (0.08) | 72.77 (0.10) | | | | |
| FSP | 70.11 (0.16) | 71.89 (0.11) | N/A | ONE | 71.41 (0.22) | 73.25 (0.13) | N/A |
| SP | 70.04 (0.21) | 72.69 (0.41) | 72.43 (0.27) | | | | |
| VID | 70.16 (0.39) | 72.61 (0.28) | 73.30 (0.31) | Teacher | 71.35 (0.28) (Generated) | 73.11 (0.36) (Generated) | 72.78 (0.12) (Generated) |
| PKT | 70.25 (0.04) | 72.61 (0.17) | 73.45 (0.19) | | | | |
| FT | 70.22 (0.10) | 72.37 (0.31) | 71.59 (0.15) | ECD | 70.83 (0.10) | 72.64 (0.15) | 72.75 (0.07) |
| NST | 69.53 (0.15) | 71.96 (0.07) | 72.24 (0.22) | ECD* | 71.56 (0.14) | 73.48 (0.18) | 73.78 (0.17) |
| | | | | ECD+FitNet | 71.08 (0.16) | 72.91 (0.21) | 73.07 (0.12) |
| RKD | 69.25 (0.05) | 71.82 (0.34) | 72.22 (0.20) | ECD+CRD | 71.46 (0.09) | 73.52 (0.11) | 73.49 (0.16) |
| CC | 69.48 (0.19) | 71.48 (0.21) | 72.21 (0.25) | ECD (dens) | 71.58 (0.21) | 73.43 (0.18) | 73.24 (0.11) |
| CRD | 71.46 (0.09) | 73.48 (0.13) | **74.14 (0.22)** | ECD (dens)* | **72.23 (0.19)** | **74.35 (0.16)** | 73.99 (0.15) |

**More discussion for ECD training**. As in Table 16, we record the best performance of each model during ECD and ECD* training. We can find that the performance of the teacher model is significantly improved by connected with the features of the student model under ECD training. So the student model gets more improvements. And the performance of each model is further improved in ECD* training.

Table 16: Detailed results on the CIFAR-100 dataset. [†] indicates using learning rate warm-up to smooth the training process. S represents the student model, T represents the teacher model and T+ represents the teacher model connected with the features of the student model under ECD training. We report top-1 "mean (std)" accuracies (%) over 3 runs.

| | Vanilla | | ECD | | ECD* | | |
|---|---|---|---|---|---|---|---|
| Model | S | T | S | T+ | S | T+ | Ensemble |
| ResNet20 | 68.78 (0.22) | 71.05 (0.35) | 70.75 (0.29) | 72.40 (0.24) | 71.07 (0.13) | 73.41 (0.12) | 73.50 (0.16) |
| ResNet32 | 70.80 (0.15) | 72.78 (0.44) | 72.40 (0.14) | 74.83 (0.13) | 73.09 (0.14) | 75.28 (0.16) | 75.71 (0.07) |
| ResNet44 | 71.88 (0.13) | 73.48 (0.48) | 73.53 (0.16) | 75.34 (0.06) | 74.08 (0.51) | 76.13 (0.12) | 76.60 (0.17) |
| ResNet56 | 72.29 (0.17) | 73.79 (0.51) | 73.94 (0.08) | 75.64 (0.09) | 74.88 (0.05) | 76.06 (0.15) | 77.53 (0.23) |
| ResNet110 | 73.15 (0.45) | 75.32 (0.64) | 75.11 (0.14) | 75.89 (0.10) | 75.47 (0.56) | 78.12 (0.16) | 78.41 (0.64) |
| ResNet164 | 76.64 (0.61) | 78.78 (0.73) | 77.94 (0.39) | 79.31 (0.21) | 78.34 (0.79) | 79.73 (0.10) | 80.99 (0.17) |
| ResNet110[†] | 74.41 (0.09) | 75.68 (0.21) | 75.59 (0.10) | 76.52 (0.16) | 76.46 (0.20) | 78.09 (0.65) | 78.89 (0.12) |
| ResNet164[†] | 77.15 (0.10) | 78.47 (0.11) | 78.33 (0.12) | 80.14 (0.20) | 78.64 (0.19) | 80.07 (0.20) | 81.12 (0.13) |
| WRN-40-1 | 71.44 (0.14) | 72.48 (0.23) | 72.55 (0.17) | 72.87 (0.09) | 72.83 (0.05) | 73.52 (0.12) | 74.25 (0.15) |
| WRN-40-2 | 75.96 (0.12) | 77.23 (0.19) | 76.55 (0.15) | 78.57 (0.19) | 76.64 (0.07) | 79.12 (0.14) | 79.45 (0.17) |

**Attention map visualization**. In our ECD, the feature information of the teacher model can be transmitted to the student model through the feature connection. As shown in Figure 4, the attention map of the student network is more similar to that of the teacher network under ECD training.

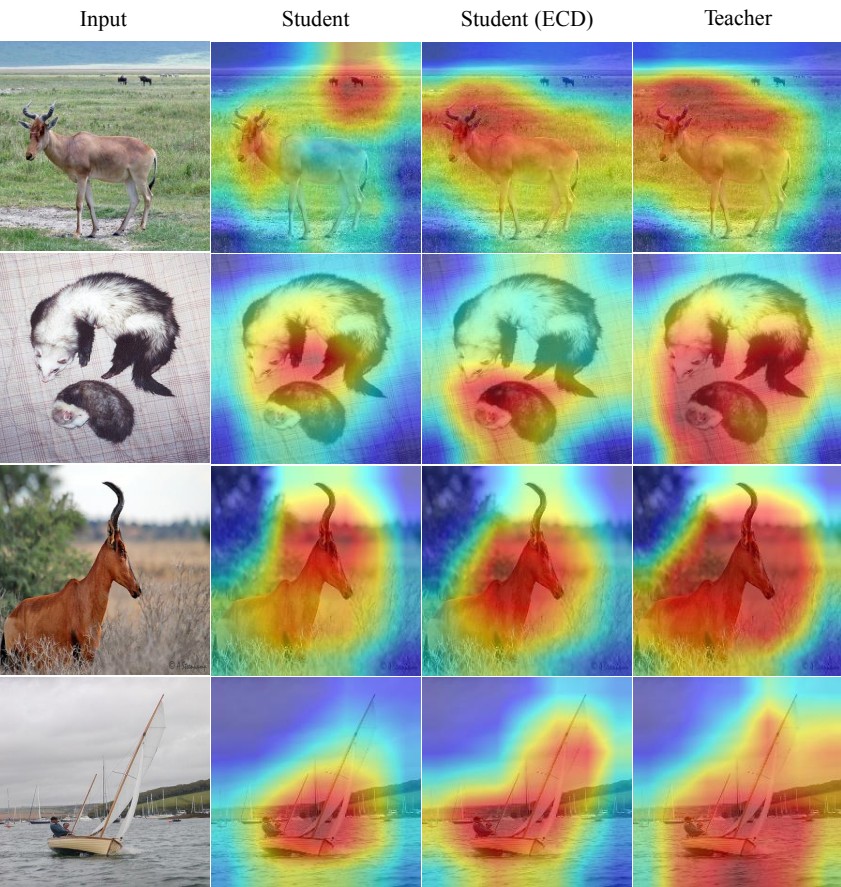

Figure 4: Comparison on the Grad-CAM++ (Chattopadhyay et al. (2018)) visualization results between the features of the student, the student with ECD connections, and the teacher network. Results are obtained based on ImageNet with ResNet18. Target objects in the upper two images are misclassified by the student network while correctly classified by the student with ECD connections.

