# OpenReview forum: "Explicit Connection Distillation"
_ICLR.cc/2021/Conference — Reject_

### Official Review · AnonReviewer3 · 2020-10-28
**Official Blind Review #3**

**Rating:** 5
**Confidence:** 5

**Review:**

Overall, I vote for marginally below the acceptance threshold. I think the idea is somewhat novel for the Explicit Connection Distillation, especially for cross-network layer-to-layer gradients propagation. This paper proposes a new strategy of Knowledge distillation called  Explicit Connection Distillation (ECD), and the ECD achieving knowledge transfer via cross-network layer-to-layer gradients propagation without considering conventional knowledge distillation losses. Experiments on popular image classification tasks show the proposed method is effective.
However, several concerns including the clarity of the paper and some additional ablation studies ( see cons below) result in the decision.

##########################################################################
Pros:
1) The knowledge distillation by cross-network layer-to-layer gradients propagation is somewhat novel to me.
2) This paper is easy to follow, the methodology part is clear to me.
3) The experiments part show detail ablation study of each component and the supplementary lists almost detail of experiments which help the community to reproduce the proposed methods.

##########################################################################
Cons:
1) The first concern is about motivation. (1): The author claims conventional KD methods leads to complex optimization objectives since they introduce two extra hyper-parameters. To my best of the knowledge, these two parameters have not too much search space. e.p Temperature is from 3-5 and the weight is T^2 from Hilton's paper and following paper.  (2): The drawback of one-stage KD methods is a little bit overclaim,  Both ONE and DML can be applied to a variety of architecture. In my opinion, the teacher design of  ECD  follows a similar strategy with ONE and its variants, which is the teacher is wider than the student.  Overall, I think the motivation of this paper needs to be very careful to clear.
2) The fairness of comparison. Is the Dynamic additive convolution component used to in student network, does this will influence the comparison of  Table2, Does ONE and DML also use that?
3) Why the automatically generated teachers of ECD is much lower than other methods in Table2 in term of performance and results in higher student performance. Is there any explanation here, like such as [1][2]?
4) Could you provide the computation cost comparison of the proposed method and other one-stage methods in Table2.
5) Some recently SOTA work is missed [3][4], although I know the performance of this paper is outperformed. I think they need to be discussed.


Reference:
[1]: Improved Knowledge Distillation via Teacher Assistant

[2]: Search to distill: pearls are everywhere but not the eyes

[3]: Online Knowledge Distillation via Collaborative Learning

[4]: Peer Collaborative Learning for Online Knowledge Distillation

---

> ### Author Response · Authors · 2020-11-25
> **Responses to the review of AnonReviewer3**
>
> Thank you so much for the constructive comments and the recognition of the novelty of our method. Please see our below responses to your questions one by one.
>
> 1.**To your first concern about the motivation** “…Overall, I think the motivation of this paper needs to be very careful to clear.”
>
> **Our responses**: Thank you for pointing out this critical problem. Accordingly, in our revised submission, we carefully rephrase the motivation of the paper, remove all inaccurate/irrelevant/misleading claims and descriptions, and strengthen the merits and connections of existing KD methods and our method. Please see **the second part** of our top-level comments titled “General Responses and the Summary of Changes in Our Revised Submission” for detailed responses.
>
> 2.**To your second concern about the fairness of comparison** “Is…does…Does ONE and DML also use that?”
>
> **Our responses**: (1) The dynamic additive convolution component is only used to the automatically generated teachers of our method, but not to the students. That is, the students keep the same to all methods in Table 2. (2) For One and DML, in our original submission, instead of using the dynamic additive convolution component we followed the original papers to set the architecture choices of the teachers given the student networks. (3) Following your suggestion, in our revised submission, under the same architecture settings of the students and teachers (using dynamic additive convolutions) we provide an experimental comparison of our method with DML and Ensemble Distillation (a simplified version of ONE) as they are all one-stage KD methods. Please see Table 8 for detailed results.
>
> 3.**To your third concern about Table 2** “Why… Is there any explanation here, like such as [1][2]?”
>
> **Our responses**: (1) In Table 2, the results of the teachers are all reported for independently trained models following the common evaluation protocol used in the KD research. For the reference methods we followed the original papers to set the architecture choices of the teachers given the student networks, while for our method the teachers (have the same depth as the students) are automatically generated from the student networks. This is the main reason why the automatically generated teachers of ECD have lower accuracy compared to the teachers (usually are much deeper than the students) used by the other methods. (2) In our revised submission, we provide the results of the automatically generated teachers of ECD and ECD* after joint training, showing improved accuracy compared to the independently trained ones. Please see Table 16 in the Appendix for detailed results. (3) For more experiments under the same architecture settings of the teacher and student networks, **please see our third response to your second concern**.
>
> 4.**To your fourth concern about the computational cost comparison** “Could you provide…in Table2?”
>
> **Our responses**: (1) In Table 2, we mainly compared the performance of our methods with both two-stage KD methods and one-stage KD methods. In the comparison, on the one side, for the architecture choices of the teacher and student networks, we used the original settings of the reference methods; on the other side, two-stage KD methods and one-stage KD methods have different training strategies (separate training vs. simultaneous training). As a result, a fair comparison of the computational cost for the training cannot be made. (2) To your request, as our method belongs to the one-stage KD solution family, in our revised submission we provide a fair comparison of our method with DML and Ensemble Distillation (a simplified version of ONE). In the comparison, we use the same architecture settings of the students and teachers (using dynamic additive convolutions). Experimental results show our method has almost the same computational cost in training, please see the caption of Table 8 for detailed comparison.
>
> 5.**To your last comment** “Some… missed [3][4]...I know the performance of this paper is outperformed…I think they need to be discussed.”
>
> **Our responses**: Thank you for pointing out these two works. We cite them in our revised submission. Actually, we noticed them before. They are one-stage KD methods, using different strategies to combine DML and ONE to achieve improved knowledge distillation performance. Specifically, [3] follows the basic framework of DML but replaces the mutual distillation (student-to-student) by the ensemble(learnt over students)-to-students distillation, and [4] follows the basic framework of ONE but jointly uses the mutual distillation (student-to-student) and the ensemble-to-students distillation.
>
> **Finally**, regarding more experiments and improvements that we have made, you are referred to our top-level comments titled “General Responses and the Summary of Changes in Our Revised Submission”, our revised submission and our responses to the other reviewers.

---

### Official Review · AnonReviewer2 · 2020-10-29
**An Interesting Knowledge Distillation Method. But I Do Not Understand Why It Works**

**Rating:** 6
**Confidence:** 3

**Review:**

The authors propose a new knowledge distillation method applicable to convolutional neural networks. Given the architecture of a student network, the teacher network is constructed by replacing each convolutional layer with a dynamic additive convolutional layer. In this way, the teacher network is guaranteed to be more capable than the student network. Then, the teacher and student models are trained together, minimizing their own training losses. In order to "distill" the student model, the student and teacher model are inter-connected in a way such that the student model receives gradient flow from the teacher network.

Pros:
1. The proposed method is straightforward and easy to implement.
2. By adding an additional distillation loss on the logits (the ECD* method), it achieved competitive distillation results.

Cons:
1. The authors claim that the gradient flow (from the teacher model to the student model) helps improving the student model. But I do not see why. The gradient is from the teacher's loss. So by applying the gradient on the student model, only the teacher's loss can be improved. This is my greatest concern. Similarly, from Equation (2), I do not see why optimizing \theta_T would help the generalization ability of the student model.
2. The authors made several claims about existing knowledge distillation methods. For me, many of the claims are not meaningful. For example: "..in real applications, for two-stage KD methods well pre-trained teacher models are usually not available, ..." This is true. But what is the drawback of "well pre-trained teacher models are usually not available"? All distillation methods require a teacher model. The proposed method still need to train a teacher model. In fact, I might claim that two-stage KD methods can utilize existing pre-trained teacher models, whereas the proposed method always has to train a teacher model during distillation. Similar claim was made on "designing complex distillation losses" as a drawback. But a user does not have to design new forms of losses if he/she stick to existing methods. On the other hand, if the proposed method became popular, researchers may add auxiliary losses on top of the proposed method, and that would not seem like a drawback for the proposed method.

Some confusions:
1. When comparing to different distillation methods in Table 2, are all teacher models have the same architecture?
2. When generating the teacher model, do the authors initialize the weights of the teacher model randomly, or according to the student model?

After reading the author feedback, I upgrade my rating to 6. See responses in this thread for reasons.

---

> ### Author Response · Authors · 2020-11-25
> **Responses to the review of AnonReviewer2**
>
> Thank you so much for the constructive comments and the recognition of the novelty of our method. Please see our below responses to your questions one by one.
>
> 1.**To your greatest concern about how/why the proposed method works** “… But I do not see why... I do not see why optimizing \theta_T would help the generalization ability of the student model.”
>
> **Our responses**: Thank you for raising this critical problem. Accordingly, we address this problem by providing an interpretation of our method in a new perspective of the deep supervision methodology, as well as some visualization results to illustrate that our method also works in a mimicking manner by the cross-network layer-to-layer gradients propagation from the teacher to the student. Please see **the first part** of our top-level comments titled “General Responses and the Summary of Changes in Our Revised Submission” for detailed responses.
>
> 2.**To your second concern about the claims of the existing KD methods** “The authors made several claims about existing knowledge distillation methods. For me, many of the claims are not meaningful...if the proposed method became popular, researchers may add auxiliary losses on top of the proposed method, and that would not seem like a drawback for the proposed method.”
>
> **Our responses**: Thank you for pointing out this important issue. Accordingly, in our revised submission, we carefully rephrase the motivation of the paper, remove all inaccurate/irrelevant/misleading claims and discussions, and strengthen the merits and connections of different KD methods. Please see **the second part** of our top-level comments titled “General Responses and the Summary of Changes in Our Revised Submission” for detailed responses.
>
> 3.**To your question** “When comparing to different distillation methods in Table 2, are all teacher models have the same architecture?”
>
> **Our responses**: (1) In Table 2, the students are the same to all methods, and for the reference methods we followed the original papers to set the architecture choices of the teachers (usually are much deeper than the students) given the student networks, while for our method the teachers (have the same depth as the students) are automatically generated from the student networks. As a result, for two-stage KD methods the teacher models have the same architecture, while for one-stage KD methods including our method the teacher architectures are different. (2) In Table 2, the results of the teachers are all reported for independently trained models following the common evaluation protocol used in the KD research. (3) Even under (1) and (2), you may notice the automatically generated teachers of our method have lower accuracy compared to the teachers used by the other methods, but the student models of our method usually have better accuracy. (4) In our revised submission, we provide the results of the automatically generated teachers of ECD and ECD* after joint training, showing improved accuracy compared to the independently trained ones. Please see Table 16 in the Appendix for detailed results. (5) In our revised submission, we also provide an experimental comparison of our method with DML and Ensemble Distillation (a simplified version of ONE) as they are all one-stage KD methods, under the same architecture settings of the students and teachers (using dynamic additive convolutions). Please see Table 8 in the Appendix for detailed results.
>
> 4.**To your question** “When generating the teacher model, do the authors initialize the weights of the teacher model randomly, or according to the student model?”
>
> **Our responses**: The weights of both the automatically generated teacher model and its corresponding student model are all initialized randomly. That is, in our method, besides the structural feature alignment to the given student at each convolutional layer by using dynamic additive convolutions instead of standard convolutions, no other change is made to the teacher both in architecture and training.
>
> **Finally**, regarding more experiments and improvements that we have made, you are referred to our top-level comments titled “General Responses and the Summary of Changes in Our Revised Submission”, our revised submission and our responses to the other reviewers.

---

### Official Review · AnonReviewer4 · 2020-10-29
**Interesting and novel work knowledge distillation, with some unanswered concerns**

**Rating:** 7
**Confidence:** 4

**Review:**

Summary:

The paper proposes new KD framework, i.e., Explicit Connection Distillation (ECD), which unlike existing methods, designs teacher network that is well aligned with the student architecture and trains both the networks simultaneously using explicit dense feature connections. The proposed method is evaluated on CIFAR-100 and ImageNet datasets.

Strengths:

- The proposed method neither requires any explicit pre-trained teacher network, nor any distillation loss. So, the method overcomes the problem of selecting teacher network or alternatives of distillation losses for the task at hand.
- By design, the generated teacher network has features aligned with the student network at every layer.

Concerns:

- Though existing works involves complex optimization in terms of losses but the hyperparameters involved in distillation like the weight on distillation loss or the temperature value is not so sensitive like learning rate. Even without careful tuning, decent distillation performance can be achieved with moderate temperature, high weight on distillation loss and low weight on cross entropy loss. So, this is not a major limitation in existing methods.
- In the proposed ECD framework, both the teacher and student networks are trained simultaneously, so number of trainable parameters (teacher parameters + student parameters) would be large. So, the method may not work well in case of limited amount of training samples.
- Selecting an optimal value of kernel number ‘n’ is a concern.
- The gain in performance in Table 1 for WRN-40-2 is marginal. So, it seems the proposed method may not be effective on some architectures like wide ResNets where the network channels are widened.
- In Table 5, marginal improvement using ECD over stage wise feature supervision.
- Table 4 shows shallow layers migrate more information than higher layers and dense connections are preferred on shallow layers only to get optimal performance. But identifying the layer from which high level semantics would be captured is non-trivial.

Queries for authors:

- Any restriction or range of values that alpha can take?
- All the experiments are done with n=16, how the performance changes by varying ‘n’?
- Is the performance of the teacher reported in Table 1, obtained through auxiliary teacher involving feature connections with the student network?
- While training using the proposed ECD, how to decide number of epochs for training (based on either teacher or student performance on validation data)?
- Details about ECD* and how learnable ensemble is applied is not mentioned in detail even in Appendix.

General Remarks:

While the creation of auxiliary teacher directly from the student network removes its dependencies from pre-trained teacher but dependency on several design choices like the number of dynamic additive convolutions for the first module and appropriate places for adding connection paths in the second module for explicit flow of layer to layer gradients remain.

---

> ### Author Response · Authors · 2020-11-25
> **Responses to the review of AnonReviewer4: Part 2**
>
> 6.**To your last concern about** “Table 4 shows shallow layers migrate more information than higher layers and dense connections are preferred on shallow layers only to get optimal performance. But identifying the layer from which high level semantics would be captured is non-trivial.”
>
> **Our responses**: You are correct. In Table 4, we studied the influence of the locations to add dense feature connections. We were also somewhat surprised by the final finding that shallow features are more useful than deeper features (having increased semantics) in improving the training of the student by our method. Indeed, it is non-trivial to identify the optimal locations of the layers, but our current choice generalizes well across different networks and across different datasets as validated by our experiments. The prevailing automatic search technique maybe an alternative solution to this problem, but it needs to perform the searching for each student separately.
>
> 7.**To your first question** “Any restriction or range of values that alpha can take?”
> ** Our responses**: For dynamic additive convolutions, the set of alpha values are usually learnt with a softmax function or a sigmoid function conditioned on the dimension-reduced version (a vector) of the input feature channels (a tensor). We used a softmax function, thus each alpha is within [0, 1] and their sum equals to 1. We clarify this in our revised submission.
>
> 8.**To your second question** “All the experiments are done with n=16, how the performance changes by varying ‘n’?”
>
> **Our responses**:  Please see **our responses to your third concern**.
>
> 9.**To your third question** “Is the performance of the teacher reported in Table 1, obtained through auxiliary teacher involving feature connections with the student network?”
>
> ** Our responses**: (1) In Table 1, the results of the teachers are all reported for independently trained models following the common evaluation protocol used in the KD research. This also applies to the other Tables including the reference methods reported in Table 2. (2) In our revised submission, we also provide the results of the automatically generated teachers of ECD and ECD* after joint training, showing improved accuracy compared to the independently trained ones. Please see Table 16 in the Appendix for detailed results. (3) For more experiments under the same architecture settings of the teacher and student networks, please see **our responses to the second concern from reviewer 3**.
>
> 10.**To your fourth question** “While training using the proposed ECD, how to decide number of epochs for training (based on either teacher or student performance on validation data)?”
>
> **Our responses**: (1) As the student and the teacher are from the same CNN backbone but with different depth/width, for a fair comparison we followed the standard training settings (200 epochs on CIFAR-100) of the student and teacher networks in all experiments. (2) In our revised submission, we also provide experiments trained with an increased number of training epochs (240 epochs), showing slightly better results for all methods and similar conclusions. Please see Table 15 in the Appendix for detailed experiments. (3) In our revised submission, we also provide an experimental comparison of our method with DML and Ensemble Distillation (a simplified version of ONE) as they are all one-stage KD methods, under the same architecture settings of the students and teachers (using dynamic additive convolutions). Please see Table 8 for detailed experiments.
>
> 11.**To your last question** “Details about ECD* and how learnable ensemble is applied is not mentioned in detail even in Appendix.”
>
> **Our responses**: (1) In our original submission, ECD* was clarified in Section 4 of the main body “Based on ECD, we also apply another ensemble distillation on the outputted logits from the two heads of the merged model to further boost the performance of each head, which is named as ECD*”. (2) More specifically, the ensemble teacher is learnt as a weighted sum of the teacher and student predications (in logits), and the weighted predications are used to guide the training of the teacher as well as the student.
>
> 12.**To your general remarks** “While…removes its dependencies from pre-trained teacher but dependency on several design choices like the number of dynamic additive convolutions for the first module and appropriate places for adding connection paths in the second module...”
>
> **Our responses**: You are referred to **our responses to your concerns, especially the third and last concerns**.
>
> **Finally**, regarding more experiments and improvements that we have made, you are referred to our top-level comments titled “General Responses and the Summary of Changes in Our Revised Submission”, our revised submission and our responses to the other reviewers.

---

> ### Author Response · Authors · 2020-11-25
> **Responses to the review of AnonReviewer4: Part 1**
>
> Thank you so much for the constructive comments and the recognition of the novelty of our method. Please see our below responses to your concerns and questions one by one.
>
> 1.**To your first concern about the limitations of the existing KD methods** “Though existing works involves complex optimization in terms of losses...this is not a major limitation in existing methods.”
>
> **Our responses**: Thank you for pointing out this critical problem. Accordingly, in our revised submission, we carefully rephrase the motivation of the paper, remove all inaccurate/irrelevant/misleading claims and discussions, and strengthen the merits and connections of different KD methods. Please see **the second part** of our top-level comments titled “General Responses and the Summary of Changes in Our Revised Submission” for detailed responses.
>
> 2.**To your second concern about the limitations of the proposed methods** “In the proposed ECD… number of trainable parameters (teacher parameters + student parameters) would be large. So, the method may not work well in case of limited amount of training samples.”
>
> **Our responses**: You are mostly correct. This is a common problem to one-stage KD methods including our method as they need to train both the student and teacher (peers or the ensemble of multiple peers or multi-branch heads) models from scratch simultaneously, introducing more trainable parameters compared to two-stage KD methods in which the parameters of teacher model are pre-trained and fixed. In case of limited amount of training samples, besides one-stage KD methods including our method, two-stage KD methods may also not work well if the teacher model is pre-trained on the same dataset but not on another large amount of training samples.
>
> 3.**To your third concern about** “Selecting an optimal value of kernel number ‘n’ is a concern.”
>
> **Our responses**: (1) As we stated in Section 3.3 of our original submission, “n is the kernel number (we set it to 8/16 as suggested in (Yang et al., 2019)”. (2) Following your request, in our revised submission, we provide an ablative study (please see Table 13 of the Appendix) of selecting an optimal value of the kernel number n, showing the same conclusion as reported in Yang et al.’s paper.
>
> 4.**To your fourth concern about** “The gain in performance in Table 1 for WRN-40-2 is marginal. So, it seems the proposed method may not be effective on some architectures like wide ResNets where the network channels are widened.”
>
> **Our responses**: (1) Yes, this is a limitation of our method. In our original submission, we had already discussed this in Section 4.1 “…The performance is not that strong for WRN-40-2, as this architecture has already widened the network channels.” As you know, in our method we use dynamic additive convolutions when automatically generating the teacher model from the student. Consequently, the better performance of our teacher to the student is from widening the groups of convolutional kernels at each layer from 1 to n, which is overlapped with the design principle of wide ResNets, limiting the accuracy improvement space of the student model to some extent. (2) However, with our method on WRN-40-2, further improved student models can be attained by enabling the **more dense** cross-network multi-layer-to-single-layer connections/gradients propagation.
>
> |     model   |    student     |        ECD       |      ECD*      |   ECD(dens) | ECD(dens*) |
>
> | WRN-40-2 |  75.96(0.12) |  76.55(0.15) | 76.64(0.07) |  76.97(0.13) |   77.21(0.15) |
>
> 5.**To your fifth concern about** “In Table 5, marginal improvement using ECD over stage wise feature supervision.”
>
> ** Our responses**: (1) The purpose of Table 5 is to compare our ECD with different feature supervisions under the same training settings (using dynamic additive convolutions, etc.). Although the improvement of ECD over the stage-wise feature supervision is not large, but it shows the potential of our method without using distillation losses. (2) In our revised submission, we further provide experiments to show our ECD can be combined with the stage-wise feature supervision components (such as intermediate feature representations in FitNet and intermediate feature correlations in CRD), achieving improved knowledge distillation performance. Please see Table 14 and Table 15 in the Appendix for more experiments.

---

### Official Review · AnonReviewer1 · 2020-11-01
**Intriguing method but needs more work before publication**

**Rating:** 5
**Confidence:** 4

**Review:**

This paper suggests an interesting approach to knowledge distillation, which uses architectural properties rather than the loss function to encourage knowledge transfer between a teacher and a student. A student and teacher net are jointly trained on a prediction task, with forward connections from layers in the student net to layers in the teacher net. At test time, the teacher net can be stripped away. The method results in improvements in student performance compared to training the student without the teacher.

The main positives I see in this paper are:
+ modest improvements over baselines in the one-stage KD setting
+ provides a different perspective on KD compared to recent works that are based on loss functions

Although the results look promising, I think the paper is not yet ready for publication. The main issues I see are:
- no comparisons against network pruning and compression methods, which are arguably more relevant than KD baselines
- little insight or analysis of why the method works
- concerns with the experiments, detailed below

This paper positions itself as a KD method, and argues that a novelty is in doing KD via architectural tricks rather than via a loss function. This may indeed be a new perspective for the KD literature, but it isn’t without precedent. Methods that use architectural scaffolding at training time, which is removed at test time, do exist, simply under other names such as weight pruning or model compression (many of these papers are cited in the intro of the current submission). The current paper needs to make it clearer how the proposed method relates to those methods, and how it goes beyond them. Ideally this should include quantitative comparisons, or an explanation for why the other methods are not applicable.

My second major concern is that this paper provides very little in the way of an explanation for why the method works. There is an intriguing statement that the method “enhances the learning performance of the student model due to the backward gradient flows from the teacher,”  but nothing to back this statement up. Some analysis of how these gradients achieve desirable effects would greatly strengthen the paper.

My third concern is with the experiments. First, the numbers in Table 2 are somewhat lower than those reported in prior papers (see Table 1 of Tian et al. 2020). Appendix A2 states that the code from Tian et al. 2020 was used to run the comparisons. Why then the discrepancy in performance? Second, many of the prior KD methods perform best when their objective is combined with the original Hinton KD objective (see Table 7 of Tian et al. 2020), but this comparison is not provided in the current paper. These two concerns mean that I’m not sure the proposed method is really outperforming competitive baselines from prior work. Lastly, I did not find enough details about ECD* to be able to really evaluate if those results are strong or interesting.

Stylistically, I think the paper would be improved by adopting a more even-handed tone. Statements like “compared to existing KD methods, ECD is more friendly to end users thanks to its simplicity” or that the method is “simple and neat” come across more as advertisement rather than as scientific analysis. The intro argues that the one-stage nature of the proposed approach makes it more applicable than two-stage approaches, but I would say one and two-stage methods are simply targeting qualitatively different applications. Two stage approaches are useful when you are _given_ a big model, which maybe you do not have the resources or data to train, and want to compress it or adapt it, e.g. for mobile deployment. One stage approaches are useful when you are able to train the big model yourself. There are interesting tradeoffs between these two paradigms and one is not better than the other. The current paper should acknowledge this. In general, the advantages and disadvantages of the method should be discussed equally. I also think the paper overemphasizes how simple the method is. I don’t personally feel this method is any simpler than methods that use loss functions, and in fact I find the proposed method more conceptually complex since I don’t know why it works.

Despite these criticisms, something interesting does seem to be going on with this method, and I encourage the authors to pursue it further.

Minor comments:
1. Abstract: “temporally” —> “temporarily”?
2. Table 1: what is the "Baseline"? The student?
3. Table 2: citations should be added for each method

---

> ### Author Response · Authors · 2020-11-25
> **Responses to the review of AnonReviewer1**
>
> Thank you so much for the constructive comments, and the recognition of the novelty of our method. Please see our below responses to your concerns one by one.
>
> 1.**To your first concern about the baselines and quantitative comparisons with network pruning and compression methods** “This…as a KD method…why…not applicable.”
>
> **Our responses**: Our method differs from network compression methods like pruning and quantization both in the focus, formulation and application. (1) In the focus, our method intends to improve the accuracy of a given student network whose architecture is known and fixed during training, while for network pruning or quantization methods their goal is to get a slimmed or low-bit version of the given network without serious accuracy loss. (2) In the formulation, the teacher of our method is temporarily generated conditioned on strengthening feature representations of basic convolutions of the student network via replacing them with dynamic additive convolutions and keeping the other layers unchanged in structure, which makes a perfect feature alignment (both in input and output dimensions) to the student at every convolutional layer and **serves as the base** to achieve improved training of the student by the dense cross-network layer-to-layer gradients propagation. **To the best of our knowledge, there is no network pruning or quantization work sharing the similar design as ours, including the papers cited in the Introduction Section of our original submission**. (3) In the application, it is known that network pruning or quantization methods usually suffer from noticeable accuracy drop when the compression ratio is large or the bit width is small. KD based methods [1-2] can be combined with them to alleviate this problem. Because of the above facts, we argue that quantitative comparisons of our method with network pruning and quantization methods have no grounding point, and our current comparisons are reasonable and decent.
>
> 2.**To your second concern about little insight or analysis of why the method works** “…why the method…Some analysis… greatly strengthen the paper.”
>
> **Our responses**: Thank you for raising this critical problem. Accordingly, we address this problem by providing an interpretation of our method in a new perspective of the deep supervision methodology, as well as some visualization results to illustrate that our method also works in a mimicking manner by the cross-network layer-to-layer gradients propagation from the teacher to the student. Please see **the first part** of our top-level comments titled “General Responses and the Summary of Changes in Our Revised Submission” for detailed responses.
>
> 3.**To your third concern with the experiments** “First, the numbers in Table 2 are somewhat lower... Second… Lastly, I did not find enough details about ECD*…”
>
> **Our responses**: (1) As the student and the teacher are from the same CNN backbone but with different depth/width, for a fair comparison we followed the standard settings of training epochs in all experiments, using 200 epochs (please see Appendix A.1.1) instead of 240 epochs as in Tian et al. 2020. This leads to slightly decreased results to some methods. (2) In our revised submission, we also provide the comparison under the training with 240 epochs, see Table 15. In addition, we also provide the comparison of combing different KD methods with the original Hinton KD objective, see Table 14 and Table 15. (3) You are referred to **our responses to the last question of reviewer 4** for more details of ECD*.
>
> 4.**To your concern about the claims and discussions of existing KD methods and our method** “…I think the paper would be improved by adopting a more even-handed tone...”
>
> **Our responses**: Thank you for pointing out this critical issue. Accordingly, in our revised submission, we carefully rephrase the motivation of the paper, remove all inaccurate/irrelevant/misleading claims and discussions, and strengthen the merits and connections of existing KD methods and our method. Please see **the second part** of our top-level comments titled “General Responses and the Summary of Changes in Our Revised Submission” for detailed responses.
>
> 5.**Our responses to your three minor comments**, (1) Typo “temporally” is corrected. (2) Yes, the "Baseline" refers to the student. (3) In our original submission, the citations for the reference methods were put in Table 11 of the Appendix due to the limited page space.
>
> **Finally**, regarding more experiments and improvements that we have made, you are referred to our top-level comments titled “General Responses and the Summary of Changes in Our Revised Submission”, our revised submission and our responses to the other reviewers.
>
> [1] Asit Mishra, et al., Apprentice: Using Knowledge Distillation Techniques To Improve Low-Precision Network Accuracy, ICLR 2018.
>
> [2] Kakeru Mitsuno, et al., Channel Planting for Deep Neural Networks using Knowledge Distillation, ICPR 2020.

---

### Author Response · Authors · 2020-11-17
**Rebuttal and paper revision are in preparation**

We sincerely appreciate all reviewers for their thorough and constructive comments. We are really happy that the novelty of the proposed method was recognized by all reviewers. To address the concerns and requests from reviewers, we are carefully improving the explanations, claims, experiments, and discussions. Our detailed rebuttal responses, as well as a paper revision will be submitted in the following week.

We thank all reviewers again for their time, feedbacks and patience.

---

### Author Response · Authors · 2020-11-25
**General Responses and the Summary of Changes in Our Revised Submission: Part 2**

3.**To the experiments**, we made following improvements: (1) We provide an ablative study of the selection of kernel number ‘n’ in dynamic additive convolutions used to generate our teacher networks, showing the same conclusion as reported in Yang et al.’s paper, see Table 13. (2) We provide the results of the automatically generated teachers of ECD and ECD* after joint training, showing improved accuracy compared to the independently trained ones, see Table 16. (3) We provide the comparison of combing different KD methods with the original Hinton KD loss, showing improved performance for almost all methods, see Table 14. (4) We provide the comparison with an increased number of training epochs (240 epochs as in Tian et al.’s paper instead of standard 200 epochs), showing slightly better results for all methods and similar conclusions, see Table 15. (5) We provide an experimental comparison of our method with DML and Ensemble Distillation (a simplified version of ONE) as they are all one-stage KD methods, under the same architecture settings of the students and the teachers (using dynamic additive convolutions), see Table 8. (6) We provide experiments to show our ECD can be combined with the two-stage distillation losses (in FitNet and CRD) as well as the one-stage distillation losses (in DML and ONE), achieving improved knowledge distillation performance, see Table 14, Table 15 and Table 8. (7) We provide experiments to show much better student models can be attained by enabling the more dense cross-network multi-layer-to-single-layer gradients propagation in our method, please see Table 14 and Table 15. (8) We provide some visualization results to illustrate how our method works, see Figure 2 and Figure 4.

**Finally**, we sincerely hope our above general responses together with our detailed responses to each of you are helpful to address your concerns, questions and requests.

[1] Chen-Yu Lee, et al., Deeply-Supervised Nets, AISTATS, 2015.

[2] Gao Huang, et al., Multi-Scale Dense Networks for Resource Efficient Image Classification, ICLR 2018.

[3] Zhenli Zhang, et al., ExFuse: Enhancing Feature Fusion for Semantic Segmentation, ECCV 2018.

[4] Dawei Sun, et al., Deeply-Supervised Knowledge Synergy, CVPR 2019.

---

### Author Response · Authors · 2020-11-25
**General Responses and the Summary of Changes in Our Revised Submission: Part 1**

To all reviewers,

We sincerely thank you all for thorough and constructive comments, and the recognition of the novelty of our work. In the past two weeks, we carefully improved the explanations, the claims/discussions, and the experiments to address all your concerns and requests. Here, we summarize our responses to the common concerns, and the changes in our revised submission.

1.**To the explanation of why our method works**, our responses are: (1) From the optimization objective of Equation (2), a common understanding of our method is the joint optimization will improve the training of the teacher as the student is merged into the teacher by the dense layer-to-layer feature aggregation progressively. It is true the teacher model is consistently improved as can be seen from the new results in Table 16, but comparatively the improvement to the student model is usually larger. We can explain this in a reverse thinking: during the joint training, note that the student and the teacher are merged into one single network by dense layer-to-layer feature connections from the student to the teacher, that means during the inference the teacher will depend on the student but the student does not depend on the teacher (stripping away the teacher from the student). **In such a perspective, by Equation (2), the teacher can be naturally treated as the auxiliary supervision to the student, which is well in line with the deep supervision (DS) methodology [1] in terms of both the mathematical formulation and the inference execution**. In the DS methodology, it directly uses individual auxiliary supervisions added to several intermediate layers of a CNN model to ease gradients propagation, and they are discarded during inference. However, they usually bring marginal improvement on modern CNNs as reported [2-4]. In a sharp contrast to existing DS methods, in our ECD, the teacher acting as the auxiliary supervision is not only based on its own structure generated with a perfect feature alignment (both in input and output dimensions) to the student at every convolutional layers, but also is based on the dense layer-to-layer feature aggregation from the student to the teacher progressively, enabling dense backward layer-to-layer gradients propagation from the teacher to the student and boosting the training of the student. Therefore, our method extends the deep supervision methodology in a new perspective on developing KD research. (2) Furthermore, we also provide experiments to show even better student models can be attained by enabling more dense feature connections (extending layer-to-layer feature connections at the same depth to at the different depths) in our method, please see Table 14 and Table 15. (3) Empirically, we provide some visualization results (by CAM++ and TSNE tools) to illustrate that our method works in a mimicking manner just like existing KD methods to some extent, see Figure 2 and Figure 4.

2.**To the claims of existing KD methods, the motivation of our method, and their connections**, we made following improvements: (1) To the claims of existing KD methods, we remove all inaccurate/irrelevant/misleading descriptions (limitations like “complex optimization objectives”, “careful tuning of each hyper-parameter”, “commercially unattractive”, etc.). (2) To the motivation of our method, we make it more clear: unlike existing KD methods that are based on distillation losses, we investigate a different technical perspective on KD design, using dense feature connections from a given student to its well-aligned generated teacher to merge them into one network temporarily, and achieving the knowledge transfer goal by the layer-to-layer gradients distillation from the teacher to the student via joint training from scratch. (3) Prevailing distillation losses can be readily combined into our framework to get improved KD performance in an online manner, as validated by extensive experiments (see Table 8, Table 14 and Table 15).

---

### Decision · Program_Chairs · 2021-01-07
**Final Decision**

**Decision:**

Reject

**Comment:**

Knowledge distillation (KD) has been widely used in practice for deployment.  In this paper, a variant of KD is proposed: given a student network, an auxiliary teacher architecture is temporarily generated via dynamic additive convolutions; dense feature connections are introduced to co-train the teacher and student models. The proposed method is novel and interesting. Empirical results showed that the proposed method can perform better than several KD variants.  However, it is unclear why the proposed method works, although the authors tried to address this issue in their rebuttal.   Besides this,  a bigger concern on this work is that it missed a comparison with a recent approach in [1] which looks much simpler and performs significantly better on similar experiments.  In [1], their ResNet50 (0.5x) is smaller than the student model in this paper (which used more filters on the top) but showed much stronger performance on both relative and absolute improvements over the same baseline (training from scratch) for the ImageNet classification task. On the technical side, the method in [1] simply uses the original ResNet50 as the teacher model,  and the student model ResNet50 (0.5x) progressively mimics the intermediate outputs of the teacher model from layer to layer. [1] also contains a  theoretic analysis  (mean-field analysis based) to support their method. Comparing with the method in [1], the proposed method here is more complicated, less motivated, and less efficient.

[1] D. Zhou, M. Ye, C. Chen, T. Meng, M. Tan, X. Song, Q. Le, Q. Liu and D. Schuurmans. Go Wide, Then Narrow: Efficient Training of Deep Thin Networks. ICML 2020.